# Use of Urinary Biomarkers in Discriminating Interstitial Cystitis/Bladder Pain Syndrome from Male Lower Urinary Tract Dysfunctions

**DOI:** 10.3390/ijms241512055

**Published:** 2023-07-27

**Authors:** Wan-Ru Yu, Yuan-Hong Jiang, Jia-Fong Jhang, Hann-Chorng Kuo

**Affiliations:** 1Department of Nursing, Hualien Tzu Chi Hospital, Buddhist Tzu Chi General Hospital, Hualien 970, Taiwan; wanzu666@gmail.com; 2Institute of Medical Sciences, Tzu Chi University, Hualien 970, Taiwan; 3Department of Urology, Hualien Tzu Chi Hospital and Tzu Chi University, Buddhist Tzu Chi Medical Foundation, 707, Section 3, Chung Yang Road, Hualien 970, Taiwan; redeemerhd@gmail.com (Y.-H.J.); alur1984@hotmail.com (J.-F.J.)

**Keywords:** urine biomarker, lower urinary-tract symptoms, detrusor overactivity, hypersensitive bladder, interstitial cystitis, bladder dysfunction, bladder-outlet obstruction, lower urinary-tract symptoms, overactive bladder, urinary biomarkers

## Abstract

To analyze the urinary biomarkers in men with lower urinary-tract symptoms (LUTS) and identify interstitial cystitis/bladder pain syndrome (IC/BPS) from the other lower urinary-tract dysfunctions (LUTDs) by the levels of characteristic urinary biomarkers. In total, 198 men with LUTS were prospectively enrolled and urine samples were collected before intervention or medical treatment. Videourodynamic studies were routinely performed and the LUTDs were diagnosed as having bladder-outlet obstruction (BOO) such as bladder-neck dysfunction, benign prostatic obstruction, or poor relaxation of external sphincter (PRES); and bladder dysfunction such as detrusor overactivity (DO), hypersensitive bladder (HSB), and IC/BPS. Patients suspicious of IC/BPS were further confirmed by cystoscopic hydrodistention under anesthesia. The urine samples were investigated for 11 urinary inflammatory biomarkers including eotaxin, IL-6, IL-8, CXCL10, MCP-1, MIP-1β, RANTES, TNF-α, NGF, BDNF, and PGE2; and 3 oxidative stress biomarkers 8-OHdG, 8-isoprostane, and TAC. The urinary biomarker levels were analyzed between LUTD subgroups and IC/BPS patients. The results of this study revealed that among the patients, IC/BPS was diagnosed in 48, BOO in 66, DO in 25, HSB in 27, PRES in 15, and normal in 17. Patients with BOO had a higher detrusor pressure and BOO index than IC/BPS, whereas patients with IC/BPS, BOO, and DO had a smaller cystometric bladder capacity than the PRES and normal subgroups. Among the urinary biomarkers, patients with IC/BPS had significantly higher levels of eotaxin, MCP-1, TNF-α, 8-OHdG, and TAC than all other LUTD subgroups. By a combination of different characteristic urinary biomarkers, TNF-α, and eotaxin, either alone or in combination, had the highest sensitivity, specificity, positive predictive value, and negative predictive value to discriminate IC/BPS from patients of all other LUTD subgroups, BOO, DO, or HSB subgroups. Inflammatory biomarker MCP-1 and oxidative stress biomarkers 8-OHdG and TAC, although significantly higher in IC/BPS than normal and PRES subgroups, did not have a diagnostic value between male patients with IC/BPS and the BOO, DO, or HSB subgroups. The study concluded that using urinary TNF-α and eotaxin levels, either alone or in combination, can be used as biomarkers to discriminate patients with IC/BPS from the other LUTD subgroups in men with LUTS.

## 1. Introduction

Male lower urinary-tract symptoms (LUTS) include storage, voiding, and postvoid symptoms. The storage symptoms include frequent urination, urgency, urgency incontinence, nocturia, and, sometimes, bladder pain at a full bladder. Male patients might complain of several kinds of LUTS in these three categories [1]. The International Prostate Symptom Score (IPSS) has been widely applied to assess the severity of male LUTS; however, the IPSS could not provide diagnostic value for a specific lower urinary-tract dysfunction (LUTD) [2]. Men with LUTS are usually considered as having bladder-outlet obstruction (BOO) including bladder-neck dysfunction (BND) and benign prostatic obstruction [3]. Over half of men with LUTS might also result from hypersensitive bladder (HSB), overactive bladder (OAB), underactive bladder, or urethral sphincter dysfunction [4]. Among male patients with storage LUTS involving bladder pain who do not respond to conventional medications for BOO or bladder dysfunction, interstitial cystitis/bladder pain syndrome (IC/BPS) should be considered [5].

For precision diagnosis of different LUTDs in men with LUTS, a pressure-flow study to demonstrate the presence of BOO is important. A pressure-flow study provides valuable information on detrusor function and impaired contractility in patients with or without BOO [6]. Further, in combination with video, it is possible to differentiate BPO, BND, urethral sphincter dysfunction, and bladder dysfunction in male LUTS [7]. In the International Consultation of Incontinence report, the committee recommended that a pressure-flow study or videourodynamic study was recommended to be performed before an invasive procedure is planned to treat male LUTS suggestive of BOO [8].

In clinical practice, a certain portion of men may report LUTS and painful complaints at the bladder, perineum, testis, or scrotum without a remarkable anatomical BOO. Some patients will be treated as chronic prostatitis/chronic pelvic pain syndrome (CP/CPPS) if they have LUTS and painful symptoms but the response to monomodal therapy is generally poor [9]. Although the incidence of IC/BPS in male LUTDs is not high, patients should be carefully diagnosed and treated with specific therapies for IC/BPS [10]. Men with IC-like LUTS diagnosed as IC/BPS may also have bladder-outlet dysfunction as well as bladder dysfunction, causing a hypersensitive and painful bladder [11]. Since patients with IC/BPS usually urinate at a small volume, difficulty in initiation, slow stream, and incomplete voiding are frequent complaints, mimicking a lower urinary-tract condition of BOO. In real-world practice, IC/BPS diagnosis in men is more difficult than CP/CPPS because IC/BPS was traditionally considered a female disease [12]. The IC-like symptoms in patients with LUTD might, in part, originate from bladder-outlet dysfunction rather than the bladder alone [9,13]. With a videourodynamic study, we still cannot identify IC/BPS in patients who are predominantly bothered with storage symptoms. An accurate diagnosis of IC/BPS depends on cystoscopic hydrodistention and a bladder biopsy under anesthesia [14].

Behind the LUTD, the molecular pathophysiology involves chronic inflammation and bladder fibrosis due to BOO [15], increased oxidative stress due to high intravesical pressure [16], and increased different subtypes of sensory afferents causing bladder hypersensitivity or overactivity [17]. Urinary proteins have been considered to represent the condition of kidney diseases and reflect the bladder conditions after BOO [18,19]. A previous study found urinary levels of epidermal growth factor, matrix-metalloproteinase-1, interleukin-6 (IL-6), nerve growth factor (NGF), and monocyte-chemoattractant protein-1 (MCP-1), representing inflammation and tissue remodeling, can be used to predict bladder dysfunction in men with LUTS and persistent postoperative detrusor overactivity (DO) [20]. In addition, urinary NGF, brain-derived neurotrophic factor (BDNF), and prostaglandin E2 (PGE2) are increased in many OAB patients and these biomarkers can help identify OAB phenotypes [21,22]. Our recent study on female patients with IC/BPS revealed that IC/BPS patients had significantly higher urinary MCP-1, eotaxin, tumor necrosis factor (TNF) -α, PGE2, 8-hydroxy-2-deoxyguanosine (8-OHdG), and 8-isoprostane levels than the controls [23]. Urinary chemokines and cytokines are significantly associated with bladder conditions and can be useful biomarkers to predict treatment outcomes of IC/BPS [24]. Urinary oxidative stress biomarkers like 8-OHdG and 8-isoprostane showed a significant diagnostic ability to distinguish the European Society for the Study of Interstitial Cystitis (ESSIC) type 2 IC/BPS from the controls [25]. However, the urinary biomarkers for the identification of IC/BPS in male patients with LUTS are still lacking. This study aims to investigate whether we can make a precision diagnosis of IC/BPS in men with LUTS based on the characteristic urinary biomarker levels.

## 2. Results

A total of 198 male patients were included in the final analysis. The age and videourodynamic parameters are shown in Table 1. Patients with IC/BPS were significantly younger than the other LUTD subgroups. After videourodynamic study and cystoscopic hydrodistention, patients were divided into six subgroups, including IC/BPS (*n* = 48), BOO (*n* = 66), DO (*n* = 25), HSB (*n* = 27), PRES (*n* = 15), and normal tracing (*n* = 17). The urodynamic parameters did not show to be highly different except for a higher voiding; Pdet and greater BOOI noted in the BOO subgroup. Compared with the normal subgroup, most LUTD subgroups had an increased FSF, FS, and small CBC, and a subnormal Qmax. (Table 1) Using ROC analysis, a CBC of ≧408.5 mL could identify patients with normal tracing or PRES, with an AUC of 0.761, sensitivity was 60.0%, specificity was 87.7%, PPV was 40.9%, and NPV was 93.9%.

The urinary biomarker concentrations in each subgroup are shown in Table 2. If we compared the urinary biomarker levels between IC/BPS and non-IC/BPS LUTD subgroups, patients with IC/BPS had a significantly higher level of eotaxin, MCP-1, TNF-α, 8-OHdG, and TAC; and a significantly lower level of CXCL10. The other urinary biomarkers did not significantly differ among subgroups such as IL-6, IL-8, MIP-1β, RANTES, PGE2, NGF, and BDNF.

Table 3 shows the AUC, COV, sensitivity, specificity, PPV, and NPV of each urinary biomarker in discriminating IC/BPS from the other LUTD subgroups. Among the biomarkers, eotaxin (≥2.290 pg/mL), TNF-α (≥1.165 pg/mL), 8-OHdG (≥126.1 ng/mL), and TAC (≥526.7 mmol/μL) had an AUC > 7.0, whereas a higher TNF-α had the best sensitivity, specificity, PPV, and NPV.

When we compare the urinary biomarkers between patients with IC/BPS and the BOO, DO, or HSB subgroup, the sensitivity, specificity, PPV, and NPV remained similar to the comparison between IC/BPS and all LUTD subgroups. Higher levels of eotaxin and TNF-α were noted to have the best PPV and NPV in discriminating patients with IC/BPS. Although 8-OHdG and TAC were also satisfactory in identifying IC/BPS from patients with BOO, DO, or HSB, the PPV and NPV were inferior to eotaxin and TNF-α (Table 4).

When we combine two or more urinary biomarkers with a more than 70% sensitivity, specificity, PPV, and NPV in discriminating IC/BPS from all LUTD subgroups, combining a higher eotaxin (≥2.290 pg/mL) and a higher TNF-α (≥1.165 pg/mL) can provide a satisfactory diagnostic value for identifying IC/BPS from men with LUTD. The sensitivity was 91.7%, specificity was 92.0%, PPV was 78.6%, and NPV was 97.2% for identifying patients with IC/BPS (Figure 1).

Based on the results of the analysis of VUDS parameters and urinary biomarker levels between patients with IC/BPS and other LUTD subgroups, it is possible to use CBC for differentiating the normal and PRES subgroups by a CBC of ≥408.5 mL. Among the patients with IC/BPS, BOO, DO, and HSB, using a higher urinary level of eotaxin (≥2.290 pg/mL) or TNF-α (≥1.165 pg/mL), most of the patients with IC/BPS can be identified from the other LUTD subgroups. (Figure 2) If we still cannot separate male patients with BOO, DO, or HSB, a videourodynamic study may be helpful in discriminating each LUTD and providing effective treatment.

## 3. Discussion

The results of this study found that patients with IC/BPS had significantly higher levels of eotaxin, MCP-1, TNF-α, 8-OHdG, and TAC than the other LUTD subgroups. By combinations of different characteristic urinary biomarkers, TNF-α, and eotaxin, whether alone or in combination, had the highest sensitivity, specificity, PPV, and NPV to discriminate IC/BPS from patients of all LUTD subgroups, the BOO, DO, or HSB subgroups. The inflammatory biomarker MCP-1 and oxidative stress biomarkers 8-OHdG and TAC, although they were significantly higher in IC/BPS than in the normal and PRES subgroups, and did not provide a diagnostic value between patients with IC/BPS and the BOO, DO, or HSB subgroups.

Bladder inflammation has been considered as the main pathogenesis of IC/BPS, resulting in bladder urothelial denudation and symptoms of bladder pain and frequency [14]. Nevertheless, the definitive etiology of IC/BPS remains unclear and a poor treatment outcome remains a challenge to urologists [5]. Chronic bladder inflammation can lead to central sensitization, resulting in sensory nerve activation and bladder hypersensitivity, which may induce afferent nerve overactivity [14,26] and, in a part of patients, cause efferent nerve activation of the bladder neck and urethral sphincter, resulting in a high rate of voiding dysfunction in patients with IC/BPS [27].

The IC-like LUTS, such as dysuria, small voided volume, and frequency urgency, are highly prevalent in male patients with IC/BPS [28]. These symptoms might lead to an incorrect diagnosis of CP/CPPS or BOO and medical treatment being given but the results are often not satisfactory [29]. In a previous functional study of male IC/BPS patients, most of the patients were found to have LUTD such as DO, BND, dysfunctional voiding, and PRES according to the precision diagnosis of a videourodynamic study. The results also reflect that bladder outlet dysfunction might coexist with (or secondary to) the bladder inflammation of IC/BPS but might not be the etiology of IC/BPS [5].

Male patients with IC-like LUTS are more commonly misdiagnosed as having anatomical BOO and surgical intervention might be mistakenly performed if the initial medical treatment targeting BOO fails. Compared with the LUTS in female IC/BPS patients, the incidences of DO, voiding dysfunctions, and urgency LUTS are significantly higher in male IC/BPS patients [5]. The higher incidence of DO and storage LUTS in male IC/BPS patients might result in an enhanced guarding effect of the bladder neck, urethral striated muscle, or pelvic floor muscles, which ultimately causes a dysfunctional bladder outlet and voiding symptoms. Therefore, identifying IC/BPS among male patients with LUTD is essential.

Our previous studies of urinary biomarkers in female patients with IC/BPS varied widely in the urinary levels of characteristic urinary biomarkers due to different patient subgroupings and disease severity. Among urinary cytokines, RANTES, MIP-1β, and IL-8 were reported to be highly sensitive, and MCP-1, CXCL10, and eotaxin were highly specific to differentiate women with IC/BPS from controls [30]. Between women with IC/BPS and OAB, MIP-1β can be the initial screening biomarker to differentiate the disease from the control groups. Then, eotaxin, CXCL10, and RANTES can be used to diagnose women with IC/BPS with a satisfactory predictive rate [31]. In female patients with dysfunctional voiding, urinary levels of IL-1β, IL-8, BDNF, and 8-OHdG were significantly higher than the controls [32]. Female patients with IC/BPS had significantly higher urinary levels of MCP-1, eotaxin, TNF-α, and PGE2 [24]. Among the women with IC/BPS, urinary levels of MIP-1β and TNF-α had the highest AUC to predict IC/BPS from the controls, whereas patients with Hunner’s IC had significantly higher urinary levels of IL-8, CXCL10, BDNF, IL-6, and RANTES than non-Hunner’s IC [33]. Further, we also found that urinary oxidative stress biomarkers such as 8-OHdG and 8-isoprostane had a high diagnostic ability to distinguish ESSIC type 2 IC/BPS from controls [25], and also have the potential to identify urodynamic DO in women with stress urinary incontinence [34]. The higher urinary inflammatory and oxidative stress biomarker levels are associated with a more severe bladder condition of IC/BPS in women [23]. The lower levels of urinary CXCL10, 8-OHdG, and 8-isoprostanese were also found to associate with a better long-term treatment outcome in women with IC/BPS [24]. All these data were obtained from women with IC/BPS or OAB; however, there is a trend that increases in urinary inflammatory and oxidative stress biomarkers are associated with IC/BPS and might predict disease severity and treatment outcome.

In the current study, male patients with IC/BPS had higher inflammatory urinary biomarkers eotaxin and TNF-α than that in patients with BOO, DO, and HSB, indicating that the IC symptoms are associated with a higher inflammatory condition of the urinary bladder. Using these two urinary biomarkers, we can identify most of the male patients with IC/BPS from the other LUTD subgroups. Although oxidative stress biomarkers are also relatively higher in IC/BPS patients than the other LUTD subgroups, the biomarkers 8-OHdG and 8-isoprostane are also elevated in patients with BOO, DO, and HSB, suggesting that oxidative stress is also commonly present in these LUTD subgroups. Therefore, the inflammatory biomarkers might have a higher diagnostic value in discriminating male patients with IC/BPS from LUTD subgroups.

The results of this study are somewhat different from that in our previous biomarker studies in female IC/BPS patients. Among the urinary biomarkers, which are higher in IC/BPS, eotaxin, MCP-1, TNF-α, 8-OHdG, and TAC were found to be significantly higher in IC/BPS patients than those in other LUTD subgroups. However, only eotaxin and TNF-α, either alone or in combination, had satisfactory predictive value in discriminating IC/BPS. TNF-α is a proinflammatory cytokine, which could cause inflammation resulting in bladder damage [35]. The bladder-tissue level of TNF-α was significantly increased in patients with Hunner’s IC. In IC/BPS patients, mast cell activation and the excessive release of TNF-α could elicit an inflammatory response; therefore, the urine level of TNF-α level could increase [36]. Eotaxin has been considered as a potential urinary biomarker in the diagnosis of patients with IC/BPS [37]. We have previously found that the urinary level of eotaxin had a high specificity for diagnosing ESSIC type 2 IC/BPS [30]. This result suggests that in male IC/BPS, the bladder inflammatory condition is significantly higher than that which occurs in male patients with BOO, DO, or HSB. Since the degree of BOO, DO, and HSB also varied widely in the enrolled patients, the levels of the other urinary inflammatory proteins and oxidative biomarkers also have a wide range and might not be useful in the diagnostic role of IC/BPS.

Among male patients with IC/BPS, a high percentage of patients had LUTD. Patients with BOO and DO might also have bladder discomfort other than urgency and dysuria. An increase in bladder inflammation could induce afferent nerve hyperactivity and result in bladder outlet and pelvic-floor muscle hypertonicity, as that noted in women with IC/BPS [38]. Therefore, clinical symptoms are not reliable to discriminate different LUTD subtypes in male patients with LUTS. Although videourodynamic study and cystoscopic hydrodistention can be used as a precision diagnosis for IC/BPS, it should be welcome if a noninvasive test like urinary biomarkers can replace these invasive procedures. The results of this study provide a simple diagnostic algorithm to discriminate IC/BPS in male patients with LUTS.

The strength of this study is, for the first time, measuring urine biomarkers in male patients with different videourodynamic diagnosed LUTD. Using higher levels of characteristic urine biomarkers such as TNF-α and eotaxin, we can identify IC/BPS and provide specific diagnosis and treatment. The limitation of the study is that urinary biomarkers are usually not stable and might be affected by several medical comorbidities. Patients with BOO and DO also have bladder inflammation and increased oxidative stress; therefore, a certain percentage of overlap might exist between male patients with IC/BPS and the other LUTDs. The urinary biomarkers investigated in this study are commonly used in laboratory research for inflammatory diseases. Some of them have already been used in routine central laboratory tests such as oxidative stress biomarkers. Therefore, in the near future, it is feasible to use urinary biomarkers as a screening test for differential diagnosis of specific diseases with LUTS.

## 4. Materials and Methods

This prospective study enrolled 198 male patients with LUTS at an outpatient clinic. All patients were investigated by the office of urological examinations, including transrectal sonography of the prostate, uroflowmetry including maximum flow rate (Qmax), voided volume, flow pattern, postvoid residual (PVR) volume, and prostate-specific antigen and 50 mL of urine were collected for urinary protein analysis. Male patients who were less than 20 years old with previous prostatic surgery and received medication for LUTS within recent 3 months, with a neurogenic lesion such as spinal cord injury, stroke, Parkinson’s disease, or dementia, having chronic urinary retention, having acute urinary tract infection, were not included in this study. In addition, patients with severe diabetes mellitus, congestive heart failure, chronic kidney disease, coronary arterial disease, or chronic obstructive pulmonary diseases that might affect the lower urinary tract function, and systemic inflammation, were not enrolled in this study. An informed consent form was obtained after 30 mL of urine for sample collection was performed. This study was approved by the institutional review board of the hospital (IRB No: 109-264-B, dated 21 December 2020). All study methods were performed in accordance with the relevant guidelines and regulations.

All patients were investigated by videourodynamic study to reveal the underlying LUTDs. The videourodynamic study was performed in a standing position. The infusion rate was set at 30 mL/min and the first sensation of filling (FSF), fullness sensation (FS), bladder compliance, cystometric bladder capacity (CBC), voiding detrusor pressure (Pdet), maximum flow rate (Qmax), voided volume, and PVR were recorded. After the study, the bladder contractility index (BCI), voiding efficiency (VE), and BOO index (BOOI) were calculated. The diagnosis of various LUTDs was in accordance with the recommendations of the International Continence Society [39]. In patients who had a CBC of less than 350 mL, a potassium chloride (KCl) test was routinely performed and patients with a positive KCl test were further admitted for cystoscopic hydrodistention and bladder biopsy under anesthesia [40]. IC/BPS was diagnosed if diffuse glomerualations developed after hydrodistention and patients were classified as different subgroups according to the maximal bladder capacity (MBC) and glomerulation grade [41]. If no glomerulation or reduced bladder capacity was noted after cystoscopic hydrodistention, the patient was diagnosed as having a hypersensitive bladder. All enrolled IC/BPS patients received cystoscopic hydrodistention and were classified as ESSIC type 1 or 2 (i.e., without or with glomerulations detected during hydrodistention) [42]. The urinary biomarkers were compared among different LUTDs, including IC/BPS, BOO, DO, HSB, PRES, and normal tracing subgroups.

Cystoscopic hydrodistention was performed under general anesthesia with an intravesical pressure of 80 cm H_2_O for 10 min. The MBC was recorded and the bladder wall was carefully examined for the formation of petechia, glomerulations, splotch hemorrhage, mucosal fissures, and Hunner’s lesion after infused fluid evacuation [14]. All patients underwent bladder biopsy to exclude carcinoma and proven chronic cystitis. The glomerulation grade was classified as suggested in the Asian IC guidelines [14]. Patients with Hunner’s lesion with or without glomerulation were classified as having ulcer-type IC/BPS. Diagnosis of IC/PBS was established based on characteristic symptoms and findings after cystoscopic hydrodistention. [43] ESSIC criteria were also used for all patients for the diagnosis of IC/BPS [42]. After the diagnosis was made, the patients received subsequent medical or surgical treatments for different LUTD.

### 4.1. Assessment of Urine Biomarker Levels

Urine samples were collected from all enrolled study patients and controls. Urine was self-voided when the subjects reported a full bladder sensation. We performed urinalysis simultaneously to confirm an infection-free status before urine samples were stored. In total, 50 mL of urine were placed on ice immediately and transferred to the laboratory for preparation. The samples were centrifuged at 1800 rpm for 10 min at 4 °C. The supernatants were separated into aliquots in 1.5-mL tubes (1 mL per tube) and stored at −80 °C. Before further analyses were performed, the frozen urine samples were centrifuged at 12,000 rpm for 20 min at 4 °C, and the supernatants were used for subsequent measurements.

### 4.2. Quantification of Inflammatory Cytokines

The inflammatory cytokines, chemokines, and neurotrophins for investigation in urine samples were similar to those in our previous study [24,25]. The targeted analytes in urine were assayed using commercially available microspheres with the Milliplex^®^ Human cytokine/chemokine magnetic bead-based panel kit (Millipore, Darmstadt, Germany). A total of 11 targeted analytes included Eotaxin, IL-6, IL-8, C-X-C motif chemokine ligand 10 (CXCL10), MCP-1, macrophage inflammatory protein (MIP)-1β, regulated upon activation, normal T-cell expressed and presumably secreted (RANTES), and TNF-α measured with the multiplex kit catalog number HCYTMAG-60K-PX30; NGF was measured with the multiplex kit catalog number HADK2MAG-61K and BDNF was measured with the multiplex kit catalog number HNDG3MAG-36K. The following laboratory procedures for the quantification of these targeted analytes were performed similarly to those in our previous study [30]. The urinary PGE2 level was measured using a high-sensitivity ELISA kit (Cayman, MI, USA), according to the manufacturer’s instructions. The detailed procedures were similar to those reported in a previous study [44].

### 4.3. Urinary Oxidative Stress Biomarker Assay

The quantifications of 8-OHdG, 8-isoprostane, and total antioxidant capacity (TAC) in urine samples were performed in accordance with the manufacturer’s instructions (8-OHdG ELISA kit, Biovision, Waltham, MA, USA, K4160-100; 8-isoprostane ELIZA kit, Enzo, Farmingdale, NY, USA, DI-900-010; and TAC Assay Kit, Abcam, Cambridge, MA, USA, ab52635). The procedures used in the urine biomarker assay were in accordance with our previous report [25]. The measurements of urine oxidative stress biomarker levels were further standardized based on urinary creatinine levels measured using a commercial kit (Enzo Life Sciences Inc., Farmingdale, NY, USA, ADI-907-030A).

### 4.4. Statistical Analysis

Continuous variables were presented as means ± standard deviations and categorical variables as numbers and percentages. Outliers were defined as values outside the range between the means ± three standard deviations for each biomarker in either the study or the control group. The urinary biomarkers data were reported in the units of pg/mL, except for ng/mL in 8-OHdG and mmol/μL in TAC. Mean differences in clinical data and the levels of urine biomarkers among IC/BPS and other LUTD subgroups were compared by one-way analysis of variance (ANOVA), and a post hoc test was performed via Bonferroni’s correction. The receiver operating characteristic (ROC) curve and area under the curve (AUC) were used to analyze the cutoff value (COV) for each urinary biomarker or combined several biomarkers in discriminating IC/BPS from different LUTD subgroups or all LUTD subgroups. The sensitivity, specificity, positive predictive value (PPV), and negative predictive value (NPV) were also calculated. All calculations were performed using SPSS Statistics for Windows, Version 20.0 (IBM Corp., Armonk, NY, USA). The difference is considered statistically significant if a *p* value is less than 0.05.

## 5. Conclusions

Male patients with IC/BPS had significantly higher levels of eotaxin, MCP-1, TNF-α, 8-OHdG, and TAC than the other LUTD subgroups. Using urinary TNF-α and the eotaxin level, either alone or in combination, can be used as biomarkers to discriminate patients with IC/BPS from the other LUTD subgroups in men with LUTS.

## Figures and Tables

**Figure 1 ijms-24-12055-f001:**
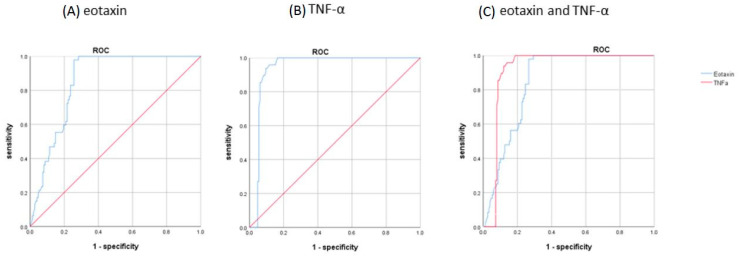
The area under the curve of (**A**) eotaxin, AUC = 0.855, (**B**) TNF-α, AUC = 0.938, and (**C**) combined eotaxin and TNF-α, sensitivity = 91.7%, specificity = 92.0%, PPV = 78.6%, NPV = 97.2%.

**Figure 2 ijms-24-12055-f002:**
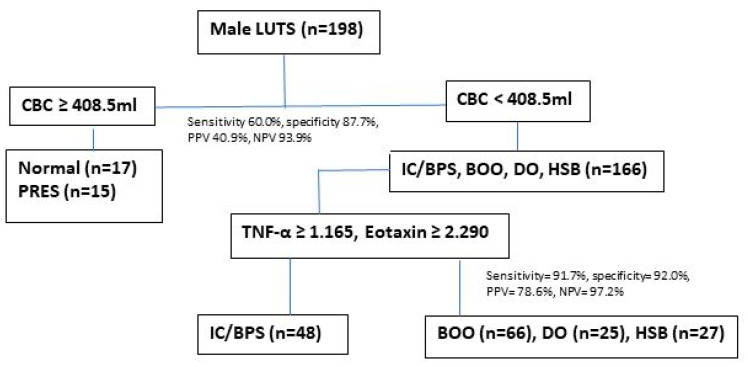
The diagnostic algorithm of discriminating IC/BPS in male patients with lower urinary-tract symptoms.

**Table 1 ijms-24-12055-t001:** Videourodynamic parameters in male patients with different lower urinary tract dysfunction.

	1. BOO (*n* = 66)	2. DO (*n* = 25)	3. HSB (*n* = 27)	4. PRES (*n* = 15)	5. Normal (*n* = 17)	6. IC/BPS (*n* = 48)	Groups 1–5 (*n* = 150)	1–5 vs. 6 *p*-Value	*p*-Value	Post Hoc
Age (years)	69.1 ± 11.7 (0)	70.6 ± 12.9 (0)	62.9 ± 16.4 (0)	64.1 ± 12.8 (0)	58.1 ± 15.1 (0)	46.9 ± 15.3 (0)	66.5 ± 13. 8 (0)	<0.001	<0.001	1234 vs. 6
Pdet (cmH_2_O)	55.3 ± 26.5 (29)	30.8 ± 14.5 (14)	33.6 ± 11.9 (16)	25.8 ± 7.2 (5)	25 ± 17.4 (12)	25.8 ± 12.4 (3)	42.4 ± 24.3 (76)	<0.001	<0.001	1 vs. 2346
Qmax (mL/s)	8.57 ± 3.1 (29)	10.8 ± 7.1 (14)	9.1 ± 5.4 (16)	8.4 ± 4.3 (5)	8.8 ± 5.5 (12)	9.0 ± 4.4 (1)	9.0 ± 4.5 (76)	0.954	0.881	
Volume (mL)	225 ± 96.7 (29)	215 ± 127 (14)	270 ± 141 (16)	274 ± 131 (5)	283 ± 185 (12)	239 ± 126 (1)	241 ± 119 (76)	0.923	0.696	
PVR (mL)	30.6 ± 47.2 (31)	7.3 ± 16.8 (14)	42 ± 95.9 (17)	145 ± 153 (7)	115 ± 170 (12)	55.4 ± 101 (1)	47.9 ± 91.4 (81)	0.681	0.140	
FSF (mL)	124.4 ± 47.8 (30)	118 ± 59.5 (14)	141 ± 61.1 (16)	161.9 ± 85.1 (5)	210 ± 57.9 (12)	126 ± 51.6 (1)	137 ± 61.7 (77)	0.326	0.015	1 vs. 45, 2 vs. 45
FS (mL)	187.4 ± 77.3 (30)	185.7 ± 94.4 (14)	240 ± 73.1 (16)	267 ± 110 (5)	325.4 ± 68.7 (12)	207 ± 86.8 (1)	215.4 ± 92.1 (77)	0.608	0.004	126 vs. 4, 126 vs. 5
Compliance	56.9 ± 57.1 (30)	72.7 ± 95.8 (14)	57.7 ± 44.6 (16)	101 ± 105 (5)	60.2 ± 22 (12)	55.7 ± 38.7 (1)	65.7 ± 69.0 (77)	0.369	0.350	
BCI	98.1 ± 27.8 (29)	84.9 ± 35.0 (14)	79.1 ± 27.0 (16)	67.8 ± 23.1 (5)	69 ± 32.6 (12)	69.8 ± 25.1 (1)	87.3 ± 30.3 (76)	0.001	<0.001	1 vs. 3456
CBC	254.3 ± 106 (29)	223 ± 129 (14)	307.9 ± 127 (16)	390 ± 142.9 (5)	398 ± 60.2 (12)	294 ± 114 (1)	286 ± 128 (76)	0.711	0.003	12 vs. 45, 4 vs. 6
cQmax	0.56 ± 0.21 (29)	0.75 ± 0.45 (14)	0.55 ± 0.26 (16)	0.45 ± 0.25 (5)	0.43 ± 0.26 (12)	0.54 ± 0.25 (2)	0.57 ± 0.28 (76)	0.605	0.235	
VE	0.9 ± 0.15 (29)	0.95 ± 0.12 (14)	0.89 ± 0.24 (16)	0.75 ± 0.29 (5)	0.7 ± 0.42 (12)	0.83 ± 0.28 (2)	0.87 ± 0.22 (76)	0.375	0.347	
BOOI	38.2 ± 28.4 (29)	9.18 ± 22.5 (14)	15.5 ± 17.6 (16)	9 ± 10.7 (5)	7.4 ± 20.6 (12)	6.6 ± 16.4 (1)	24.5 ± 27.2 (76)	<0.001	<0.001	1 vs. 2346

BOO: bladder-outlet obstruction, DO: detrusor overactivity, HSB: hypersensitive bladder, PRES: poor relaxation of external sphincter, IC/BPS: interstitial cystitis/bladder pain syndrome, LUTD: lower urinary-tract dysfunction, Pdet: detrusor pressure, Qmax: maximum flow rate, PVR: postvoid residual, FSF: first sensation of filling, FS: full sensation, BCI: bladder contractility index (= Pdet + 5 × Qmax), CBC: cystometric bladder capacity, cQmax: corrected Qmax, VE: voiding efficiency, BOOI: bladder-outlet obstruction index (= Pdet − 2 × Qmax).

**Table 2 ijms-24-12055-t002:** The urinary biomarker levels in male patients with different lower urinary-tract dysfunction.

Biomarkers *	1. BOO (*n* = 66)	2. DO (*n* = 25)	3. HSB (*n* = 27)	4. PRES (*n* = 15)	5. Normal (*n* = 17)	6. IC/BPS (*n* = 48)	Groups 1–5 (*n* = 150)	1–5 vs. 6 *p*-Value	*p*-Value	Post Hoc
Eotaxin	3.02 ± 4.54 (0)	4.2 ± 6.78 (1)	3.62 ± 4.94 (1)	1.89 ± 1.5 (0)	2.09 ± 2.61 (0)	7.99 ± 7.27 (1)	3.09 ± 4.68 (2)	<0.001	<0.001	1345 vs. 6
IL-6	1.62 ± 3.27 (0)	1.37 ± 1.63 (1)	2.88 ± 7.3 (0)	0.89 ± 0.16 (0)	0.96 ± 0.25 (0)	1.8 ± 2.37 (1)	1.66 ± 3.86 (1)	0.814	0.448	
IL-8	7.96 ± 20.1 (2)	3.23 ± 4.24 (1)	4.29 ± 8.32 (0)	2.53 ± 4.12 (0)	3.41 ± 5.19 (0)	4.06 ± 5.24 (1)	5.43 ± 14.1 (3)	0.515	0.394	
CXCL10	43.5 ± 91.8 (1)	24.2 ± 43.5 (1)	46.0 ± 109 (1)	10.6 ± 11.2 (0)	8.93 ± 12.8 (0)	8.21 ± 14.4 (1)	33.4 ± 79.2 (3)	<0.001	0.018	1 vs. 6
MCP-1	190 ± 189 (1)	244 ± 301 (1)	186 ± 191 (2)	98.0 ± 91.4 (0)	111 ± 108 (0)	272.1 ± 260 (1)	179 ± 201 (4)	0.029	0.017	45 vs. 6
MIP-1β	1.7 ± 1.92 (1)	1.3 ± 0.86 (2)	1.47 ± 1.28 (0)	1.02 ± 0.32 (0)	1.14 ± 0.78 (0)	0.9 ± 0.98 (0)	1.46 ± 1.47 (3)	0.015	0.058	
RANTES	4.78 ± 5.09 (1)	5.59 ± 5.86 (1)	5.88 ± 6.72 (1)	3.57 ± 2.14 (0)	2.88 ± 1.24 (0)	5.1 ± 5.63 (1)	4.76 ± 5.1 (3)	0.714	0.332	
PGE2	383 ± 335 (2)	505 ± 570 (0)	368 ± 299 (0)	276 ± 142 (0)	366 ± 420 (0)	405 ± 352 (0)	388 ± 377 (2)	0.780	0.558	
TNF-α	1.0 ± 0.77 (2)	0.88 ± 0.34 (1)	0.94 ± 0.57 (1)	0.84 ± 0.19 (0)	0.81 ± 0.18 (0)	1.58 ± 0.23 (0)	0.93 ± 0.59 (4)	<0.001	<0.001	12,345 vs. 6
NGF	0.18 ± 0.05 (1)	0.19 ± 0.05 (0)	0.19 ± 0.05 (1)	0.18 ± 0.07 (0)	0.16 ± 0.04 (0)	0.17 ± 0.03 (1)	0.18 ± 0.05 (2)	0.027	0.209	
BDNF	0.47 ± 0.14 (1)	0.52 ± 0.42 (0)	0.45 ± 0.16 (1)	0.56 ± 0.38 (0)	0.48 ± 0.12 (0)	0.56 ± 0.12 (1)	0.49 ± 0.24 (2)	0.037	0.160	
8-OHdG	96.0 ± 27.8 (0)	97.2 ± 31.7 (0)	98.3 ± 33.8 (0)	80.4 ± 27.7 (0)	78.0 ± 25.8 (0)	121 ± 52.8 (0)	93.0 ± 29.9 (0)	0.001	<0.001	45 vs. 6
8-isoprostane	36.5 ± 28.7 (2)	47.0 ± 47.4 (1)	55.2 ± 55.6 (0)	32.1 ± 27.4 (0)	27.7 ± 23.7 (0)	26.2 ± 31.4 (0)	40.2 ± 38.4 (3)	0.024	0.029	
TAC	355 ± 214 (0)	337 ± 249 (0)	387 ± 346 (1)	189 ± 65.3 (0)	225 ± 113 (0)	661 ± 477 (4)	326 ± 237 (1)	<0.001	<0.001	1 vs. 456, 245 vs. 6

( ) indicates outliers; * units: all data are presented by pg/mL, except for ng/mL in 8-OHdG and mmol/μL in TAC. BOO: bladder-outlet obstruction, DO: detrusor overactivity, HSB: hypersensitive bladder, PRES: poor relaxation of external sphincter, IC/BPS: interstitial cystitis/bladder pain syndrome, LUTD: lower urinary tract dysfunction, IL: interleukin, CXCL10: C-X-C motif chemokine ligand 10, MCP-1: monocyte chemoattractant protein-1, MIP: macrophage inflammatory protein, RANTES: regulated upon activation, normal T-cell expressed and presumably secreted, PG: prostaglandin, TNF: tumor necrosis factor, NGF: nerve growth factor, BDNF: nerve growth factor, 8-OHdG: 8-hydroxy-2-deoxyguanosine, TAC: total antioxidant capacity.

**Table 3 ijms-24-12055-t003:** Comparison of urinary biomarkers between patients with interstitial cystitis/bladder pain syndrome and the other lower urinary tract dysfunction subgroups.

Biomarkers *	AUC	COV *	Sensitivity	Specificity	PPV	NPV
Eotaxin	0.855	≥2.290	97.9%	74.3%	54.0%	99.1%
MCP-1	0.614	≥120.4	66.7%	54.7%	32.0%	83.7%
TNF-α	0.944	≥1.165	93.8%	88.0%	71.4%	97.8%
CXCL10	0.658	<1.715	41.7%	88.0%	52.6%	82.5%
8-OHdG	0.711	≥126.1	64.6%	86.0%	59.6%	88.4%
8-isoprostane	0.649	<12.87	45.8%	80.0%	42.3%	82.2%
TAC	0.743	≥526.7	58.3%	83.3%	52.8%	86.2%

* units: all pg/mL, except for ng/mL in 8-OHdG and mmol/μL in TAC. AUC: area under the curve, COV: cutoff value, PPV: positive predictive value, NPV: negative predictive value; Abbreviation of biomarkers: same as in footnotes of Table 2.

**Table 4 ijms-24-12055-t004:** Comparison of urinary biomarkers between patients with interstitial cystitis/bladder pain syndrome (IC/BPS) and (A) bladder-outlet obstruction (BOO), (B) detrusor overactivity (DO), and (C) hypersensitive bladder (HSB).

**(A)** **IC/BPS vs. BOO**						
**Biomarkers ***	**AUC**	**COV ***	**Sensitivity**	**Specificity**	**PPV**	**NPV**
Eotaxin	0.855	≥2.290	97.9%	75.8	74.6%	98.0
MCP-1	0.614	≥120.4	66.7%	54.5	51.6	69.2
TNF-α	0.944	≥1.165	93.8%	84.8	81.8	94.9
CXCL10	0.658	<1.715	41.7%	87.9	71.4	67.4
8-OHdG	0.711	≥126.1	64.6%	86.4	77.5	77.0
8-isoprostane	0.649	<12.87	45.8%	83.3%	66.7%	67.9%
TAC	0.743	≥526.7	58.3%	78.8%	66.7%	72.2%
**(B)** **IC/BPS vs. DO**						
**Biomarkers ***	**AUC**	**COV ***	**Sensitivity**	**Specificity**	**PPV**	**NPV**
Eotaxin	0.855	≥2.290	97.9%	64.0%	83.9%	94.1%
MCP-1	0.614	≥120.4	66.7%	48.0%	71.1%	42.9%
TNF-α	0.944	≥1.165	93.8%	88.0%	93.8%	88.0%
CXCL10	0.658	<1.715	41.7%	96.0%	95.2%	46.2%
8-OHdG	0.711	≥126.1	64.6%	80.0%	86.1%	54.1%
8-isoprostane	0.649	<12.87	45.8%	80.0%	81.5%	43.5%
TAC	0.743	≥526.7	58.3%	83.3%	84.8%	50.0%
**(C)** **IC/BPS vs. HSB**						
**Biomarkers ***	**AUC**	**COV ***	**Sensitivity**	**Specificity**	**PPV**	**NPV**
Eotaxin	0.855	≥2.290	97.9%	63.0%	82.5%	94.4%
MCP-1	0.614	≥120.4	66.7%	48.1%	69.6%	44.8%
TNF-α	0.944	≥1.165	93.8%	85.2%	91.8%	88.5%
CXCL10	0.658	<1.715	41.7%	85.2%	83.3%	45.1%
8-OHdG	0.711	≥126.1	64.6%	77.8%	83.8%	55.3%
8-isoprostane	0.649	<12.87	45.8%	81.5%	81.5%	45.8%
TAC	0.743	≥526.7	58.3%	77.8%	82.4%	51.2%

* units: all pg/mL, except for ng/mL in 8-OHdG and mmol/μL in TAC. AUC: area under the curve, COV: cutoff value, PPV: positive predictive value, NPV: negative predictive value; Abbreviation of biomarkers: same as in footnotes of Table 2.

## Data Availability

Data are available by contact with the corresponding author.

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
