# Peer review of "Use of Urinary Biomarkers in Discriminating Interstitial Cystitis/Bladder Pain Syndrome from Male Lower Urinary Tract Dysfunctions"

_ijms, 2023, doi:10.3390/ijms241512055_

Round 1

Reviewer 1 Report

we thank the authors for providing this interesting paper on the Use of Urinary Biomarkers in Discriminating Interstitial Cystitis/Bladder Pain Syndrome from Male Lower Urinary Tract Dysfunctions.

The paper is well written and methods are correct

Here some suggestions:

Strengths and limitations of the study should be disuccssed at the end of the discussion.

Table 1 is confusing for a reader, can be improved.

This is a prospective study, this is stated in the abstract but not in the main text.

There is still poor evidence on the role of these biomarkers. In the discussion please describe briefly the role of Urinary biomarkers in other urological diseases. The following paper will help.

doi: 10.23736/S2724-6051.21.04308-1

doi: 10.1007/s11934-022-01098-6

Author Response

Reviewer #1
we thank the authors for providing this interesting paper on the Use of Urinary Biomarkers in Discriminating Interstitial Cystitis/Bladder Pain Syndrome from Male Lower Urinary Tract Dysfunctions.
The paper is well written and methods are correct
Here some suggestions:
Strengths and limitations of the study should be disuccssed at the end of the discussion.
Reply: Thank you for the comment. We have added the strengths and limitations of the study in the last paragraph of the Discussion section. (Lines 407-419)
Table 1 is confusing for a reader, can be improved.
Reply: We have revised the comparison of groups in Table 1 and Table 2. LUTD has been revised to “groups 1-5”, and “LUTD vs IC” was revised to “groups 1-5 vs 6”.
This is a prospective study, this is stated in the abstract but not in the main text.
Reply: Thank you for the comment, We have added “prospective” to the method section. (Line 138)
There is still poor evidence on the role of these biomarkers. In the discussion please describe briefly the role of Urinary biomarkers in other urological diseases. The following paper will help.
Reply: Thank you for the comment. We have mentioned the role of urinary biomarkers in the other urological diseases in Introduction.  (Lines 112-133)

Reviewer 2 Report

Authors should be congratulated for their work. Male LUTS still represents a major health concern that weighs both on public health and on the state coffers. Indeed, urinary symptoms are conditions shared by several comorbidities of LUTS patients (such as OSAS, hypertension, or diabetes, PMID: 37167825). However, the authors aimed to investigate characteristic urinary biomarker levels to discriminate IC/BPS from other conditions in men with LUTS. 

The manuscript is well-written and easily readable. The methodology is robust and well-described. My main concern is on the comorbidities of LUTS patients, that were not mentioned and discussed in the current manuscript and that could hide or prevail a condition of BOO.  Moreover, it would be interesting to know the economic feasibility of urinary biomarkers, that may limit their clinical applications. 

Author Response

Reviewer #2

Authors should be congratulated for their work. Male LUTS still represents a major health concern that weighs both on public health and on the state coffers. Indeed, urinary symptoms are conditions shared by several comorbidities of LUTS patients (such as OSAS, hypertension, or diabetes, PMID: 37167825). However, the authors aimed to investigate characteristic urinary biomarker levels to discriminate IC/BPS from other conditions in men with LUTS. 
The manuscript is well-written and easily readable. The methodology is robust and well-described. My main concern is on the comorbidities of LUTS patients, that were not mentioned and discussed in the current manuscript and that could hide or prevail a condition of BOO. Moreover, it would be interesting to know the economic feasibility of urinary biomarkers, that may limit their clinical applications. 
Reply: Thank you for the comments. This study did not analyze the comorbidities of these patients. However, patients with diabetes congestive heart failure, chronic kidney disease, coronary arterial disease, or chronic obstructive pulmonary diseases that may affect the lower urinary tract function and systemic inflammation are not enrolled in this study. (Lines 146-150)
Urinary biomarkers investigated in this study are commonly used in the laboratory researches for inflammatory diseases, some of them have already been used in routine laboratory tests such as oxidative stress biomarkers. Therefore, in the near future, it is feasible to be used as a screening test for differential diagnosis of specific diseases. (Lines 414-419)